# The Impact of Social Media on Eating Disorder Risk and Self-Esteem Among Adolescents and Young Adults: A Psychosocial Analysis in Individuals Aged 16–25

**DOI:** 10.3390/nu17020219

**Published:** 2025-01-08

**Authors:** Sylwia Jaruga-Sękowska, Wiktoria Staśkiewicz-Bartecka, Joanna Woźniak-Holecka

**Affiliations:** 1Department of Health Promotion, Faculty of Public Health in Bytom, Medical University of Silesia in Katowice, ul. Piekarska 18, 41-902 Bytom, Poland; jwozniak@sum.edu.pl; 2Department of Food Technology and Quality Evaluation, Department of Dietetics, Faculty of Public Health in Bytom, Medical University of Silesia in Katowice, ul. Jordana 19, 41-808 Zabrze, Poland; wstaskiewicz@sum.edu.pl

**Keywords:** eating disorders, high school students, college student, sex, social media

## Abstract

Background/Objectives: Eating disorders (EDs) result from complex interactions of biological, psychological, social, and cultural factors, disproportionately affecting adolescents and young adults. Social media, peer pressure, and self-esteem issues contribute to ED prevalence. This study examines ED risk, eating behaviors, and self-esteem among individuals aged 16–25, exploring differences by gender, age, and social media usage. Methods: A total of 261 participants (113 high school students, 115 college students, 33 working individuals) completed the Computer-Assisted Web Interview (CAWI) between April and June 2024. This study utilized the Eating Attitudes Test-26 (EAT-26), Rosenberg Self-Esteem Scale (SES), and My Eating Habits (MEH) questionnaires. Statistical analyses, including chi-square tests and linear regression, assessed associations between ED risk, self-esteem, and social media activity. Results: ED risk was identified in 47% (n = 123) of participants, with the highest prevalence in high school students (56.6%, n = 64). Significant gender differences were observed (*p* < 0.001), with males in the high school group showing elevated ED risk (64%, n = 32). Body dissatisfaction and frequent social media comparisons increased ED risk (*p* < 0.001); 45.7% (n = 102) of participants who often compared their bodies online reported heightened risk. Photo manipulation correlated with higher ED risk (*p* = 0.005). Regression analysis revealed a significant relationship between ED risk and restrictive dieting (estimate = 0.9239; *p* < 0.001), while self-esteem had no significant effect (estimate = 0.00503, *p* = 0.977). Conclusions: This study highlights high ED risk driven by social media and body dissatisfaction. Interventions should focus on body image issues, self-acceptance, and media literacy. This study focused on a specific age group (16–25) in Poland, which may limit the ability to generalize the results to other demographic or cultural groups. Future research should include more diverse populations and objective measurements.

## 1. Introduction

Eating disorders (EDs) are the result of the interaction of many factors, which can include biological, psychological, social, and cultural factors. Each of these factors affects the individual differently at different stages of life. These factors can intermingle and exacerbate each other, making EDs complex in nature. EDs manifest themselves in the form of chronic unhealthy eating habits that negatively affect daily social and psychological life including a disturbed self-image [1]. A very high prevalence of these disorders has been noted among young people aged 15–24 [2]. Both in high school and college, students struggle with many challenges associated with a particular period of life. Environmental pressures, difficult family relationships, the influence of social media, or peer pressure can promote the development of EDs [3,4]. Untreated EDs can persist for long periods of time and are associated with poor physical and mental health [2].

Both adolescence and young adulthood are two stages of life associated with intense biological, psychological, and social changes. These life stages play a significant role in the development of an individual’s identity. Individuals at this age begin to experience emotions related to appearance which can lead to comparisons with other people or with unrealistic body models [3]. Young people, under pressure from the media and society, make decisions related to limiting their food intake by, for example, skipping meals, starving themselves, using laxatives, or excessive physical activity. This results in the development of EDs, which can seriously damage health, and their effects are associated with the highest mortality rate among mental disorders [5].

The data on the prevalence of EDs among adolescents and young adults is alarming and points to the need for in-depth research [6,7]. Adolescents and young adults with low self-esteem and high perfectionism show more severe ED symptoms. Younger participants are a particular risk group for impairment related to both anxiety/depression and EDs [8]. The media often promote unrealistic canons of beauty and figure, which can lead to exaggerated expectations of appearance and negatively affect self-esteem. In particular, young people who are under intense pressure to meet the standards of a slim figure presented in the media may feel an even greater need to maintain an ideal appearance. Studies indicate that comparing oneself to body images promoted in social media leads to an increase in dissatisfaction with one’s own appearance [9,10].

The purpose of this study is to examine the prevalence of ED risk, eating behaviors, and self-esteem among 16–25-year-olds. The study aims to identify differences in ED risk levels between the study groups, taking into account the influence of gender, age, and extent of social media use. It is expected that high school students, especially females, and those with greater exposure to social media will show a higher risk of EDs compared with students. In addition, the control group (working people) will show a lower prevalence of ED risk than both high school and college students. Given the rapid growth of social media and its tremendous influence on young people, it seems necessary to assess not only whether the mere presence of SM affects young people’s self-esteem but also what behaviors are initiated by it. Social media puts users into a constant cycle of comparing their bodies with those of other people. Increased exposure to retouched photos will disrupt perceptions of one’s own body, and the obsession with controlling one’s weight will increase. Fear of being judged can provoke behaviors aimed at manipulating one’s own appearance, and consequently lead to undesirable health behaviors that ruin physical and mental health. The growth of social media today is associated with an exceptionally strong source of pressure and social norms regarding appearance, which poses a serious threat to people’s health, creating a sense of not being pretty or fit enough.

## 2. Materials and Methods

### 2.1. Procedure of the Study

The survey was administered between April 2024 and June 2024, targeting individuals aged 16 to 25. Data collection was carried out using the Computer-Assisted Web Interviewing (CAWI) method, employing an online questionnaire. This approach is recognized as a valid and reliable method within psychological research for obtaining self-reported data. Participants were provided with detailed information regarding this study’s objectives, the assurance of anonymity, and instructions for completing the questionnaire. At the commencement of the questionnaire, information pertaining to informed and voluntary participation was clearly presented. Participants gave their consent to engage in the study, thereby retaining the right to withdraw from participation at any stage without the obligation to provide a justification and without incurring any adverse consequences. This study adhered to ethical principles, following the World Medical Association’s Declaration of Helsinki. Approval was obtained from the Bioethics Committee of the Silesian Medical University in Katowice (approval identifier BNW/NWN/0052/KB254/23, dated 20 November 2023), in accordance with the Act of 5 December 1996 on the Profession of Physician and Dentist. This ensured compliance with both international and local ethical standards.

### 2.2. Participants

In this study, the sampling methodology was developed to ensure that the collected data meets the required level of precision and reliability. This ensured that a representative group of survey participants was obtained, assuming a confidence level of 90%, a margin of error of 5% (0.05), and a fraction size of 0.5. The sample size was calculated using the standard formula for estimating proportions in a population. By adhering to this methodology, the sample design ensures that the results are statistically robust and suitable for drawing conclusions about the target population.

In total, 275 people participated in the survey; the response rate was 96%. The questionnaire was correctly completed by 261 subjects and the analysis was based on these responses; 14 questionnaires were rejected (incomplete). The inclusion criteria for the study group were as follows: (1) voluntary participation in the research with full completion of the questionnaire, and (2) an age range of 16 to 25 years. On the other hand, the exclusion criteria were a declaration of diagnosed mental or neurological disorders that may affect the reliable completion of the questionnaire, and age <16 and >26 years. The study group consisted of high school students, college students, and working people aged 16–25 years (19.6 ± 2.97), including 179 (68.6%) women and 82 (31.4%) men. The study group included 113 (43.3%) high school students, 115 (44.1%) college students, and 33 working people (12.6%).

### 2.3. Survey Tools

This study was conducted using a survey questionnaire comprising several sections. The first section collected demographic and personal information, including age, height, weight, presence of chronic illnesses (e.g., mental health conditions such as depression or ED), current medications, occupational or educational status (e.g., high school students; college student, working people), sources of nutritional knowledge, and activity of survey people in social media. Additionally, the survey incorporated standardized instruments, including the Eating Attitudes Test (EAT-26), the Rosenberg Self-Esteem Scale (SES), and the My Eating Habits (MEH) questionnaire to assess the variables of interest comprehensively.

#### 2.3.1. Eating Attitudes Test-26

The Eating Attitudes Test-26 (EAT-26), developed by Garner and Garfinkel, is a standardized instrument designed to assess attitudes and behaviors related to eating. EAT-26 is among the most widely utilized diagnostic tools in global studies on the prevalence of EDs. Specifically, it has proven to be an effective screening tool for assessing “eating disorder risk” in populations such as high school and college students [11,12]. The questionnaire comprises three subscales: (1) Dieting, (2) Bulimia and Food Preoccupation, and (3) Oral Control. The scores for each subscale are determined by summing the responses to the items computed for that particular subscale. This study focused on the total test score consisting of 26 items that assess the frequency of specific behaviors, with responses scored on a Likert scale. For questions 1–25, the scoring system is as follows: Always = 3 points; Usually = 2 points; Often = 1 point; Other responses = 0 points. In contrast, question 26 uses reverse scoring. The final score is derived by summing the scores of items 1–26, yielding a total score ranging from 0 to 78. A score of ≥20 indicates a potential risk of developing an ED, suggesting the need for consultation with a specialist for further evaluation. Higher scores reflect more severe symptoms. The 26-item version is highly reliable and valid. The overall score gives a quick snapshot of the general attitudes of the person surveyed, making it easier to interpret and compare with the scores of other individuals or groups. The overall scale can be more representative and useful in comparative analyses. The EAT-26 total score is used in research studies because it is simple to interpret, allows rapid assessment of ED risk, and has well-developed norms for comparisons in different populations [13,14,15].

#### 2.3.2. Rosenberg Self-Esteem Scale

Global self-esteem in the participants was assessed using a questionnaire that included the Rosenberg Self-Esteem Scale (SES), as adapted into Polish by Laguna, with its validity having been established in previous studies [16,17]. The Rosenberg Self-Esteem Scale measures global self-esteem, defined as an individual’s overall attitude toward themselves, encompassing both positive and negative beliefs and perceptions. Each statement on the scale is rated on a 1 to 4 scale, ranging from strongly agree to strongly disagree. The total score, derived from the responses, reflects the individual’s level of self-esteem. Scores range from 10 to 40, with the following classifications: 10–27 points indicating low self-esteem, 28–32 points indicating moderate self-esteem, and 33–40 points indicating high self-esteem [18].

#### 2.3.3. My Eating Habits Questionnaire

The My Eating Habits Questionnaire (MEH) by Ogińska-Bulik et al. is used to assess eating habits (habits) in adults and older adolescents. The results of this test can be used for research to assess the determinants that lead to overweight and obesity. The questionnaire consists of 30 statements to be answered YES or NO. In statements number 2, 5, 13, 17, and 23, 1 point is awarded for the diagnostic answer “NO”, while in the remaining items, the diagnostic answer is “YES” (1 point). The MEH questionnaire identifies 3 factors: factor I habitual overeating, factor II emotional overeating, factor III restrictive dieting. A higher score indicates a greater tendency toward abnormal eating behaviors. The test allows for the differentiation between habitual and emotionally driven overeating [19].

### 2.4. Statistical Analysis

Statistical analyses were conducted using Statistica v.13.3 (StatSoft Polska Sp. z o.o, Kraków, Poland) and the R package v. 4.0.0 (2020) under the GNU GPL (The R Foundation for Statistical Computing). For quantitative data, mean values and standard deviations (X ± SD) were computed, while qualitative data were expressed as percentages. To examine associations between categorical variables, such as gender and status, Chi-squared tests of independence were conducted. These tests were also utilized to explore the relationship between the risk of EDs and variables including time spent on social media, the purpose of social media use, body image comparisons, body satisfaction, and photo modification.

The Cramér V coefficient was applied to assess the strength of the association between the EAT-26 scores and variables such as gender and participant status. This coefficient quantifies the strength of the relationship between two categorical variables, in this instance, the ED risk score. The normality of the distribution was evaluated using the Shapiro-Wilk test. A linear regression analysis was performed to examine the relationship between the EAT-26 score, which assesses the risk of EDs, and the scores from the Questionnaire My Eating Behavior and the Rosenberg Self-Esteem Scale. The results are presented as regression coefficients (estimates) with corresponding standard deviations, t-statistics, and statistical significance levels. ANCOVA analysis was used to assess the impact of gender and daily time spent on social media on EAT-26 scores.

A *p*-value < 0.05 was used as the criterion for statistical significance.

## 3. Results

### Sample Characteristics

A total of 261 individuals participated in the study, with participants categorized into three groups based on their status: high school students (n = 113), college students (n = 115), and employed individuals (n = 33). The average body weight of the study group was 64.7 ± 13.1 kg, while the average height was 1.70 ± 8.96 m. The body mass index (BMI) was calculated using the measured body weight and height, yielding a mean BMI of 22.4 ± 3.38 (Table 1).

Participants were questioned about their social media usage, body satisfaction, body shape comparisons with others, and the editing of photos on social media. Statistically significant differences were found between the groups. For further details, please refer to Table 2.

According to the EAT-26 results presented in Table 3, 47.0% of the respondents (n = 123) were classified as being at an increased risk for an ED. Significant differences were observed between the surveyed groups (*p* = 0.027) in the overall score, as well as between male participants (*p* < 0.001). Participants’ nutritional status was assessed based on body mass index (BMI). Results were interpreted according to World Health Organization (WHO) guidelines: <18.5 underweight, 18.5–24.99 normal weight, 25–29.99 overweight, >30 obese. However, no significant differences were found between the study groups concerning body mass index (Table 3).

The average score of the surveyed in test EAT-26 was 22.0 ± 9.42. Total subscale scores: (1) dieting was 12.51 ± 5.97 (2), bulimia was 4.79 ± 2.65, and (3) oral control was 4.69 ± 2.79. The differences between groups in subscale bulimia were statistically significant (*p* = 0.04). The analysis revealed that the average score on the My Eating Habits Questionnaire was 22.0 ± 9.42, while the following scores were obtained in the three factors: factor I, habitual overeating 4.26 ± 2.74; factor II, emotional overeating 4. 26 ± 2.72; and factor III, restrictive dieting 3.58 ± 2.59. Factors I (*p* = 0.036) and II (*p* = 0.002) differed significantly between groups, suggesting differences in habitual and emotional overeating behaviors of the study groups. The study group exhibited low self-esteem, with a score of 22.18 ± 3.35. The interpretation of the index is as follows: a score between 10 and 27 points indicates low self-esteem, 28 to 30 points reflects average self-esteem, and 32 to 40 points denotes high self-esteem (Table 4).

Statistically significant differences were observed in relation to ED risk (*p* < 0.001); however, no differences were found based on gender or group. The analysis did not reveal any statistically significant differences between sex and the time spent on social media concerning ED risk (*p* = 0.872). For males, the relationship between time spent on social media and ED risk was not significant (*p* = 0.718). Similarly, no significant association was found for females (*p* = 0.802). The analysis further revealed statistically significant differences in body satisfaction and ED risk by gender (*p* = 0.011). A significant relationship was observed between body satisfaction and ED risk in men (*p* < 0.045) but in women (*p* = 0.224). For high school students, a significant association was found between satisfaction with the appearance of their own body and ED risk (*p* = 0.011), but for students (*p* = 0.855) and working people (*p* = 0.193) the differences are not significant. ED risk has been shown to increase in high school students who would change several things about their appearance. The analysis demonstrated the presence of statistically significant differences for women who compare their own bodies with other people on social media and ED risk (*p* < 0.001). On the other hand, no significant association was observed between comparison behaviors and ED risk in men (*p* = 0.296). Upon analyzing the differences among the study groups, it was observed that students were significantly more likely to engage in body comparison behaviors on social media (*p* = 0.003). In contrast, for high school students (*p* = 0.9) and working people (*p* = 753), no such relationship was shown. The analysis demonstrated statistically significant differences between the manipulation of social media photos and ED risk (*p* = 0.005). Additionally, for females, reworking photos significantly affects the increased risk of EDs (*p* = 0.047), whereas for males, this factor did not show statistical significance (*p* = 0.06). Among the three study groups, only the student group showed a significant association between this factor and an increased risk of EDs (*p* = 0.02). Detailed results are presented in Table 5.

Table 6 shows the results of the linear regression analysis to examine the relationship between several predictors, such as the overall score on the My Eating Habits Questionnaire with all factors and gender, the dependent variable, and the total score obtained on the EAT-26 test. A measure of model fit (R^2^ = 0.0115) suggested that about 1.15% of the variation in the total score obtained on the EAT-26 test could be explained by these predictors, indicating a poor overall fit. Both the overall MEH score (*p* < 0.001) and factor III restrictive diet (*p* < 0.001) have a significant effect on the EAT-26 score, with dietary restrictions having the most significant effect among the data analyzed. Overall, the model highlighted the limited role of predictors related to Factors I and II, while the total score and Factor III demonstrated a stronger and more significant impact. There was no significant effect of gender (*p* = 0.967) of sex in the overall score achieved on the EAT-26 test. The very small value (−0.0542) of the coefficient suggests no practical effect of gender on the EAT-26 score.

There is a strong association between the independent variables and the dependent variable. The model explains as much as 94.3% of the variability in EAT-26 scores. This indicates an excellent fit of the model. Only 5.7% of the variability remains unexplained by the model. All three factors have a strong and significant effect on the EAT-26 score, with “Dieting” (t = 34.2) slightly having the largest effect. They are the main determinants of the EAT-26 score. Gender is statistically significant (*p* = 0.002), meaning that the predictor has an effect on the EAT-26 score, although its effect is much smaller than that of the other variables. Male gender is associated with a 0.9619-point decrease in EAT-26 score compared with female gender. The results suggest that dietary restriction, bulimia, and oral control are worth paying special attention to as key factors in the analysis of eating and body perception problems. Gender is less important but should still be included in analyses (Table 7).

The model has a very poor fit (R^2^ = 0.013), which means that the independent variables hardly explain the variation in EAT-26 scores. Both the Rosenberg Self-Esteem Scale scores and gender have no significant effect on EAT-26 scores. The coefficient values are close to zero, and *p*-values > 0.05 confirm the lack of statistical significance (Table 8).

ANCOVA analysis of the EAT-26 results showed that none of the analyzed variables had a significant effect on the eating disorder test score. The effects of gender (F = 1.68; *p* = 0.196) and time spent daily with social media (F = 0.07; *p* = 0.978) did not reach the level of statistical significance. The interaction between gender and time spent with social media was also not significant (F = 1.51; *p* = 0.213). The high value of the residuals (22,322.3) suggests significant variability unexplained by the factors analyzed (Table 9).

Figure 1 shows the relationship between SES scores and EAT-26 scores, considering participants’ nutritional status, interpreted based on their BMI. The results indicate significant differences in the relationship between self-esteem and EAT-26 scores across different BMI groups.

## 4. Discussion

The analysis of the EAT-26 questionnaire indicated that approximately 47.1% of respondents exhibit a heightened risk of developing an ED and should pursue professional evaluation for further diagnostic assessments. High school students (56.5%) represent the group with the highest risk of developing EDs. This elevated risk may stem from the fact that adolescence is a critical period for identity formation, marked by significant biological and psychological transitions. Inappropriate patterns promoted by the media, social pressure, and lack of self-acceptance predispose to EDs, which is why people of this age need health support and education. Adolescents are particularly vulnerable to societal pressures, including unrealistic beauty standards perpetuated by social media.

Data analysis revealed a correlation between gender and ED risk among the study participants. The results indicate that males exhibited a higher prevalence of elevated ED risk. Males in the high school group exhibited a higher ED risk (64%) compared with females (56.6%), contradicting traditional assumptions that EDs predominantly affect females. This may reflect shifting societal norms, where ideals of muscularity and fitness increasingly pressure young males, especially on social media. Moreover, the stigma associated with male EDs could lead to underreporting in earlier research, underscoring the importance of developing interventions that account for gender-specific needs. Despite previous studies that have provided important information on EDs, it is still crucial to monitor them in order to obtain updated data and better adapt interventions to changing conditions.

Participants dissatisfied with their bodies or who frequently compared themselves with others on social media showed significantly higher ED risk (*p* < 0.001). This study shows a link between dissatisfaction with one’s own body, comparisons with others on social media, and ED risk, an important step toward understanding how today’s digital environment can affect young people’s mental and physical health.

This study found that high school students spend significantly more time on social media, whereas students were notably more likely to express dissatisfaction with their body image and desire changes. This observation highlights the importance of monitoring nutritional health within this group.

This study revealed that 27.6% of participants who frequently edited their photos demonstrated elevated ED risk (*p* = 0.005). Editing photos may reflect an internalized pressure to conform to beauty ideals, perpetuating negative body image and disordered eating behaviors. While many of the available studies examine the impact of comparisons with idealized images of bodies on social media, the authors’ study introduces a new dimension by focusing on the practice of photo editing by users themselves. Photo editing can be seen as a form of self-pressure to conform to unrealistic beauty standards. This study emphasizes the active participation of participants in the creation of these idealized images, rather than merely passively receiving them.

Restrictive dieting was significantly associated with ED risk (estimate = 0.9239, *p* < 0.001). This finding underscores the role of dieting as both a symptom and a potential precursor to EDs. Restrictive eating behaviors often emerge as a coping mechanism for dissatisfaction with one’s body, leading to a cycle of unhealthy eating patterns. Self-esteem did not show a significant association with ED risk (*p* = 0.977), contrary to expectations. This may suggest that ED risk is driven more by external factors, such as social media and peer comparisons, rather than intrinsic perceptions of self-worth.

Notably, no significant differences were identified between body mass index and the heightened risk of developing EDs within the study group. This finding suggests that individuals of both normal and abnormal weight may be susceptible to developing EDs. Such a finding sheds new light on the underlying mechanisms of these disorders, indicating that their causes may be more related to psychological and social factors than to physical body parameters. Focusing on a broader spectrum of factors, independent of BMI, may contribute to the development of more universal and effective strategies to support people at risk for EDs.

The research conducted by Lonergan et al. sought to examine the potential association between social media usage patterns and an increased likelihood of meeting the diagnostic criteria for an ED, as well as to explore whether gender influenced this relationship. A total of 4209 Australian adolescents participated in the survey, which utilized the Eating Disorder Examination Questionnaire alongside other self-report measures. The findings revealed gender disparities in social media usage, with adolescent girls being more likely than boys to meet the diagnostic criteria for EDs. The results show that the greater the manipulation of photos by the subjects, the greater the probability of fulfilling the criteria for EDs. In addition, studies indicate that appearance-related social media behavior may indicate a risk of developing EDs [2].

Staśkiewicz et al. conducted research involving 103 women aged 18–35 (X = 21.80 ± 3.47) with the aim of assessing the risk of EDs and analyzing socio-cultural posture toward body image. According to the EAT-26 questionnaire results, it was found that 28.6% of Polish women and 33.3% of Turkish women met the criteria, suggesting a potential risk of developing EDs [20].

The study conducted by Frieiro et al. examined attitudes and behaviors associated with EDs, as well as the Rosenberg Self-Esteem Scale, among 721 students aged 12–18. The findings indicated that adolescents who extensively use social media were at a higher risk of developing EDs. Furthermore, the researchers observed that exposure to negative content, such as violence or rejection, increased the likelihood of developing EDs [21].

Piko et al. conducted a study among 261 female college students (X = 22.0 ± 2.2 years) to investigate the relationship between EDs, body image, and media-related factors. The study found that an elevated risk of developing EDs occurred in 24.1% of the female students surveyed. Factors such as dissatisfaction with one’s body, athletic tendencies, and weight reduction behaviors were found to be associated with EDs, whereas social media usage and body mass index had a more limited impact [22].

A study by Aparicio-Martinez et al. assessed the relationship between dissatisfaction with body image, eating habits, and the occurrence of EDs among 224 southern Spanish students aged 18–25 (20 ± 0.76). The results show that the ideal of a slim figure widely disseminated in social media may promote behaviors that lead to EDs (body dissatisfaction and abnormal attitudes toward eating). On top of that, researchers have noted two major ideals of beauty: the athletic ideal, increasing compulsive exercise among respondents, and the ideal of a slim figure, correlating with eating restriction [23].

Other researchers point to another serious problem among young people, which is addiction to smartphones, which are the main tool for using social media. The findings from a study involving 1112 Chinese students indicate a significant correlation between smartphone addiction and EDs, eating habits, and lifestyle. This connection is associated with an elevated risk of developing depression and anxiety, as well as sleep disturbances, increased consumption of highly processed foods, and reduced physical activity [24].

The results of the self-reported research regarding the participants’ self-esteem show no differences between the study groups or any association with EDs. Self-esteem is an indicator that is related to self-perception, general well-being, and eating behaviors [25,26]. This factor appears to be particularly important in adolescence due to its correlations with body image assessment, and consequently, the greater the self-esteem of young people, the better the psychological adjustment that mediates the prevention of risky behaviors mediated by EDs [2,27,28,29,30], which should not be underestimated.

A study conducted by Pop et al. explored the connection between factors such as body image satisfaction, self-esteem, and feelings of loneliness among young individuals who use social media. The overall self-esteem score was notably higher in males compared with females, with male students perceiving their mental health as being in better condition than their female counterparts. It was also shown that diminished self-esteem was linked with body dissatisfaction. In addition, levels of stress, depression, and loneliness have been shown to be predictors of self-esteem [31].

Many studies in the existing literature explore the association between social media use and self-esteem. A study by Colak et al. involving 204 high school students (age X = 15.9 ± 1.2) revealed that higher levels of social media addiction were negatively correlated with self-esteem and image of own body in the participants [32]. Conversely, a study by Chen et al. investigated the association between self-esteem, social media usage, and attitudes toward plastic surgery. The findings suggest that engagement with specific social media platforms and photo-editing applications is positively associated with greater acceptance of plastic surgery [33].

In addition, the study revealed that social media use can influence the risk of EDs and eating behaviors in young people. Adolescents and young adults engaging with social media tend to experience body dissatisfaction and are more prone to making comparisons with others online. This indicates that educators, healthcare providers, and other professionals should be mindful of the effects of social media on the mental and emotional well-being of individuals aged 16–25, which may necessitate the implementation of social media management strategies targeting this age group [34,35]. In general, it is crucial to design preventive strategies that focus on self-esteem and its connection to EDs, incorporating the influence of social media as a key factor [36,37]. Prevention should include the creation of informational and educational materials on the influence of social media on self-esteem in the context of EDs, which could be implemented as part of training in schools and universities [38,39].

### Strengths and Limitations

It is important to highlight that this is the first study to conduct a comprehensive analysis of the risk of EDs within the population of Polish high school and college students, as well as a control group of employed individuals aged 16–25 years. The originality of our study stems from the incorporation and in-depth analysis of variables that have not been previously explored in relation to ED risk among Polish adolescents and young adults. This study is the first in the literature to explore the relationship between the risk of EDs and subjective comparisons of one’s body with images shared on social media, body satisfaction, and the practice of photo modification on these platforms. This study examined variables associated with the impact of social media, offering an important contribution to research on external factors that could elevate the risk of EDs among young people. Additionally, this study incorporated further in-depth analyses, such as a linear regression examining the relationship between ED risk and scores from the My Eating Habits Questionnaire and the Rosenberg Self-Esteem Scale, underscoring the innovative aspect of this work. Our findings provide novel insights into the role these factors play in influencing ED risk, positioning this study as a meaningful contribution to the literature on the mental and physical health of young people.

A limitation of this study is the dependence on self-reported survey data, which may be subject to recall bias or deliberate misrepresentation by participants, particularly when dealing with sensitive issues such as EDs. It is possible that some responses are distorted by a reluctance to disclose certain information or memory errors, limiting the objectivity of the results. Another limitation of the study is the absence of objective laboratory measurements, which could have yielded important data on the potential physiological effects of EDs in young people, including their effects on overall fitness and, in particular, body weight, muscle mass, and percent body fat. The use of more sophisticated diagnostic tools would also allow confirmation or verification of the results reported by participants. A final limitation is the survey’s focus on the Polish population only, which may limit the generalizability of the results to other social or cultural groups. In the context of behavior, attitudes, or beliefs, there is a possibility that the results obtained in Poland may not be fully representative of other nations or cultures due to differences in upbringing, social values, or cultural norms. Therefore, the results of this study may be limited in terms of broad international application.

Future research should integrate more accurate physiological evaluations, such as body composition analysis, to quantify fat mass and lean mass. This would offer a more comprehensive and nuanced understanding of the physiological implications of EDs, surpassing the limitations of using BMI as the sole metric. Additionally, this study focused on a specific age group (16–25) in Poland, which may limit the ability to generalize the results to other demographic or cultural groups. Consideration should be given to including more diverse samples and using objective measurements.

## 5. Conclusions

The findings of this study emphasized the crucial need to identify risk factors related to the development of EDs in individuals aged 16–25. The results reveal that young people, particularly males, exhibited heightened susceptibility to EDs. Furthermore, the study highlighted the significant influence of social media on body image perceptions. Men and high school students exhibit greater dissatisfaction with their own body appearance, whereas women and college students are more inclined to compare their body shape with images of others.

These findings highlight the necessity for tailored interventions that focus on the mental health of young people and foster self-acceptance, which is often undermined by pressures from social media. Educational institutions should implement psychological support programs designed to reduce the focus on idealized body standards and encourage the adoption of healthy eating habits. Developing targeted interventions for younger individuals is equally crucial, as this study suggests that high school students experience distinct pressures and risks associated with body image and EDs. For young people, support should primarily aim at reducing the pressures linked to societal beauty ideals while also incorporating education on self-acceptance and promoting healthy eating behaviors.

## Figures and Tables

**Figure 1 nutrients-17-00219-f001:**
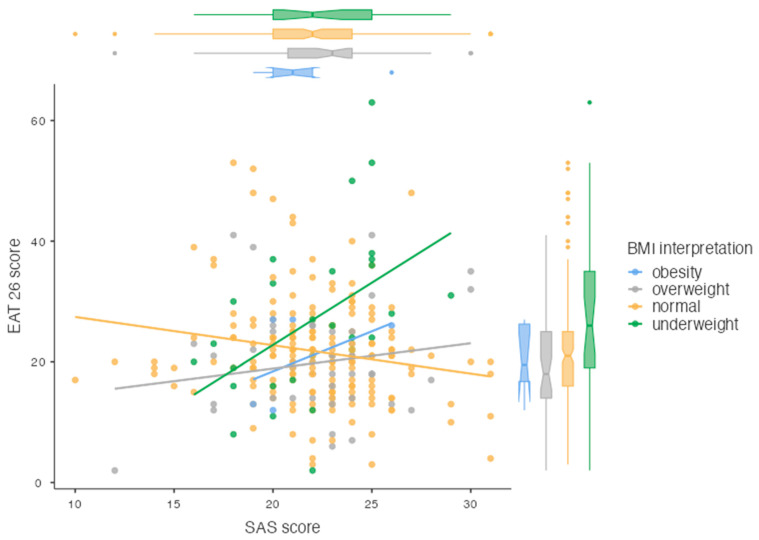
Scatter plot of SES and EAT-26 scores with nutritional status interpreted by the subjects’ BMI.

**Table 1 nutrients-17-00219-t001:** Structure of the study group.

		Total (n = 261)	HSS (n = 113)	CS (n = 115)	WP (n = 33)	*p*-Value
Gender n (%)	Female	179 (68.6)	63 (35.2)	94 (52.5)	22 (12.3)	<0.001 *
	Male	82 (31.4)	50 (61)	21 (25.6)	11 (13.4)	
Age [years] (X ± SD)		19.6 ± 2.97	16.7 ± 1.11	21.9 ± 1.53	21.8 ± 2.44	<0.001 *
Q_1_–Q_3_		19.2–20.0	16.5–16.9	21.6–22.1	20.7–22.6	
Height [cm] (X ± SD)		170 ± 8.96	171.0 ± 8.22	168 ± 8.86	171.9 ± 10.7	0.006 *
Q_1_–Q_3_		168–171	169–172	166–169	167–169	
Body mass [kg] (X ± SD)		64.7 ± 13.1	65.4 ± 13.35	62.5 ± 11.56	70.2 ± 15.81	0.016 *
Q_1_–Q_3_		63.1–66.3	63.0–68.0	60.6–65.2	63.5–73.6	
BMI [kg/m^2^] (X ± SD)		22.4 ± 3.38	22.2 ± 3.24	22.2 ± 3.44	23.5 ± 3.53	0.078
Q_1_–Q_3_		22.0–22.8	21.6–22.8	21.7–22.9	22.0–24.4	

HSS—high school students; CS—college student; WP—working people; Q_1_–Q_3_—first and third quartile; * *p* < 0.05; X—average; SD—standard deviation.

**Table 2 nutrients-17-00219-t002:** Activity of survey people in social media (n = 261).

	Total (n = 261)	HSS (n = 113)	CS (n = 115)	WP (n = 33)	*p*-Value	*V-Cramer*
Reasons for using social media. n (%)						
To relax	98 (37.5)	31 (27.4)	51 (44.3)	16 (48.5)	<0.001 *	0.222
I observe what my friends do	26 (10.0)	12 (10.6)	12 (10.4)	2 (6.1)		
I search for the daily news	99 (37.9)	61 (54)	29 (25.2)	9 (27.3)		
I search for information on nutrition	38 (14.6)	9 (8)	23 (20)	6 (18.2)		
Duration of social media use throughout the day. n (%)						
up to 1 h	27 (10.3)	9 (8.0)	12 (10.4)	6 (18.2)	0.038 *	0.160
1–2 h	71 (27.2)	23 (20.4)	40 (34.8)	8 (24.2)		
2–3 h	77 (29.5)	36 (31.9)	35 (30.4)	6 (18.2)		
above 3 h	86 (33.0)	45 (39.8)	28 (24.3)	13 (38.4)		
Satisfaction with the appearance of own body n (%)						
I am dissatisfied and would like to change many things	97 (37.2)	27 (27.8)	56 (57.7)	14 (36.4)	0.003 *	0.177
Yes, but I would change a few things in my appearance	142 (54.4)	76 (53.5)	49 (34.5)	17 (12.0)		
Yes, I would not change anything	22 (8.4)	10 (45.5)	10 (45.5)	2 (9.1)		
Do you compare your figure to other people’s photos on social media? n (%)						
Yes, often	102 (39.1)	39 (38.2)	55 (53.9)	8 (7.8)	0.018 *	0.151
Yes, sometimes	101 (38.7)	47 (46.5)	42 (41.6)	12 (11.9)		
No, never	58 (22.2)	27 (46,6)	18 (31)	13 (22.4)		
Reworking social media photos. n (%)						
Yes, always	72 (27.6)	39 (54.2)	24 (33.3)	9 (12.5)	0.05 *	1.77
Yes, sometimes	106 (40.6)	43 (40.6)	54 (50.9)	9 (8.5)		
No, never	83 (31.8)	31 (37.3)	37 (44.6)	15 (18.1)		

HSS—high school students; CS—college students; WP—working people; * *p* < 0.05.

**Table 3 nutrients-17-00219-t003:** Summary of ED risk estimation (EAT, n = 261).

EAT-26	Total (n = 261)		HSS (n = 113)		CS (n = 115)		WP (n = 33)		*p*-Value	*V-Cramer*
	No Risk	Elevated Risk	No Risk	Elevated Risk	No Risk	Elevated Risk	No Risk	Elevated Risk		
Gender. n (%)										
Total (n = 261)	138 (53)	123 (47)	49 (43.3)	64 (56.6)	69 (60)	46 (40)	20 (60.6)	13 (39.4)	0.027 *	0.1665
M (n = 82)	43 (52)	39 (48)	18 (36)	32 (64)	16 (76.2)	5 (23.8)	9 (81.8)	2 (18.2)	<0.001 *	0.4128
F (n = 179)	95 (53)	84 (49)	31 (49.2)	32 (59.8)	53 (56.4)	41 (43.6)	11 (50)	11 (50)	0.646	0.0699
Body Mass Index. n (%)										
Underweight (n = 29)	20 (69)	9 (31)	10 (66.7)	5 (33,3)	9 (75)	3 (25)	1 (50)	1 (50)	0.749	0.141
Normal (n = 180)	96 (53)	84 (47)	34 (44.2)	43 (55.8)	49 (59.8)	33 (40.2)	13 (61.9)	8 (38.1)	0.101	0.16
Overweight (n = 44)	19 (43)	25 (57)	4 (22.2)	14 (77.8)	10 (58,8)	7 (41.2)	5 (55.6)	4 (44.4)	0.065	0.353
Obesity (n = 8)	3 (38)	5 (63)	1 (33.3)	2 (66.7)	1 (25)	3 (75)	1 (100)	0	0.376	0.494

HSS—high school students; CS—college student; WP—working people; M—male; F—female; * *p* < 0.05.

**Table 4 nutrients-17-00219-t004:** Summary of Eating Attitudes Test-26, My Eating Habits, and Rosenberg Self-Esteem Scale results (n = 261).

Variable	Total (n = 261) (X ± SD)	Q_1_–Q_3_	HSS (n = 113) (X ± SD)	Q_1_–Q_3_	CS (n = 115) (X ± SD)	Q_1_–Q_3_	WP (n = 33) (X ± SD)	Q_1_–Q_3_	*p*-Value
EAT-26 Total	22.0 ± 9.42	20.8–23.1	21.1 ± 10.3	19.1–23.0	22.4 ± 8.9	20.8–24.1	23.5 ± 7.85	20.7–26.3	0.067
Dieting	12.51 ± 5.97	11.8–13.2	12.19 ± 6.42	11.0–13.4	12.54 ± 5.86	11.5–13.6	13.48 ± 4.6	11.9–15.1	0.171
Bulimia	4.79 ± 2.65	4.46–5.11	4.3 ± 2.59	3.82–4.78	5.1 ± 2.6	4.62–5.59	5.33 ± 2.8	4.34–6.33	0.04 *
Oral Control	4.69 ± 2.79	4.36–5.01	4.57 ± 3.1	3.99–5.14	4.8 ± 2.48	4.34–5.26	4.7 ± 1.91	4.02–5.37	0.322
My Eating Habits	12.1 ± 6.59	11.3–12.9	10.85 ± 5.35	9.85–11.8	13.3 ± 7.22	12.0–14.6	12.21 ± 7.47	9.56–14.9	0.065
Factor I	4.26 ± 2.74	3.9304.6	3.72 ± 2.4	3.26–4.16	4.68 ± 2.89	4.14–5.21	4.73 ± 3.03	3.65–5.8	0.036 *
Factor II	4.26 ± 2.72	3.93–4.59	3.62 ± 2.43	3.17–4.07	4.95 ± 2.80	4.43–5.47	4.06 ± 2.86	3.05–5.07	0.002 *
Factor III	3.58 ± 2.59	3.26–3.89	3.52 ± 2.31	3.09–3.95	3.68 ± 2.84	3.15–4.2	3.42 ± 2.62	2.49–4.35	0.938
Rosenberg Self-Esteem Scale	22.18 ± 3.35	21.8–22.6	22.3 ± 3.55	21.6–23.0	22.0 ± 3.16	21.5–22.6	22.2 ± 3.36	21.0–23.4	0.752

HSS—high school students; CS—college student; WP—working people; Q_1_–Q_3_—first and third quartile; * *p* < 0.05; X—average; SD—standard deviation.

**Table 5 nutrients-17-00219-t005:** The assessment of ED risk was conducted using the EAT-26 scale, considering factors such as social media usage, comparisons of body shape with others on social media, body satisfaction in the study group, and reworking photos (n = 261).

	Total (n = 261)		Male		Female		HSS (n = 113)		CS (n = 115)		WP (n = 33)	
	No Risk	Risk	No Risk	Risk	No Risk	Risk	No Risk	Risk	No Risk	Risk	No Risk	Risk
Reasons for using social media. n (%)												
To relax	50 (36.2)	47 (38.2)	15 (34.9)	9 (23.1)	36 (37.9)	38 (45.2)	14 (28.6)	17 (26.6)	27 (39.1)	22 (47.8)	9 (45)	7 (53.8)
I observe what my friends do	13 (9.4)	13 (10.6)	5 (11.6)	4 (10.3)	8 (8.4)	9 (10.7)	5 (10.2)	7 (10.9)	7 (10.1)	5 (10.9)	1 (5)	1 (7.7)
I search for the daily news	50 (36.2)	49 (39.8)	21 (48.8)	23 (59)	29 (30.5)	26 (31)	26 (53.1)	35 (54.7)	17 (24.6)	12 (26.1)	7 (35)	2 (15.4)
I search for information on nutrition	24 (17.4)	14 (11.4)	2 (4.7)	3 (7.7)	22 (23.2)	11 (13.1)	4 (8.2)	5 (7.8)	17 (24.6)	6 (13)	3 (15)	3 (23.1)
*p*-value	<0.001 *		0.352		0.634		0.995		0.485		0.656	
*V-Cramer*	0.583		0.1354		0.1444		0.0254		0.1458		0.2213	
Duration of social media use throughout the day. n (%)												
up to 1 h	16 (11.6)	11 (8.9)	6 (14)	4 (10.3)	10 (10.5)	7 (8.3)	4 (8.2)	5 (7.8)	8 (11.6)	4 (8.7)	4 (20)	2 (15.4)
1–2 h	39 (28.3)	32 (26)	12 (27.9)	8 (20.5)	27 (28.4)	24 (28.6)	8 (16.3)	15 (23.4)	27 (39.1)	13 (28.3)	4 (20)	4 (30.8)
2–3 h	38 (27.5)	39 (31.7)	11 (25.6)	10 (25.6)	27 (28.4)	29 (34.5)	13 (26.5)	23 (35.9)	20 (29)	15 (32.6)	5 (25)	1 (7.6)
above 3 h	45 (32.6)	41 (33.3)	14 (32.5)	17 (43.6)	31 (32.6)	24 (28.6)	24 (49)	21 (32.8)	14 (20.3)	14 (30.2)	7 (25)	6 (46.2)
*p*-value	0.812		0.718		0.802		0.351		0.485		0.569	
*V-Cramer*	0.060		0.128		0.075		0.171		0.146		0.247	
Satisfaction with the appearance of own body n (%)												
I am dissatisfied and would like to change many things	63 (45.7)	34 (27.6)	15 (34.9)	7 (17.9)	48 (50.5)	27 (32.1)	18 (36.7)	9 (14.1)	35 (50.7)	21 (45.7)	10 (50)	4 (30.8)
Yes, but I would change a few things in my appearance	65 (47.1)	77 (62.6)	22 (51.2)	25 (64.1)	43 (45.3)	52 (61.9)	29 (59.2)	47 (73.4)	28 (40.6)	21 (45.7)	8 (40)	9 (69.2)
Yes, I would not change anything	10 (7.2)	12 (9.8)	6 (14)	7 (17.9)	4 (4.2)	5 (6)	2 (4.1)	8 (12.5)	6 (8.7)	4 (8.7)	2 (10)	0
*p*-value	0.011 *		0.045 *		0.224		0.011 *		0.855		0.193	
*V-Cramer*	0.186		0.186		0.191		0.283		0.052		0.316	
Do you compare your figure to other people’s photos on social media? n (%)												
Yes, often	63 (45.7)	39 (31.7)	7 (16.3)	12 (30.8)	56 (58.9)	27 (32.1)	18 (36.7)	21 (32.8)	41 (59.5)	14 (30.4)	4 (20)	4 (30.8)
Yes, sometimes	45 (32.6)	56 (45.4)	18 (41.9)	14 (35.9)	27 (28.4)	42 (50)	20 (40.8)	27 (42.2)	17 (24.6)	25 (54.3)	8 (40)	4 (30.8)
No, never	30 (21.7)	28 (22.8)	18 (41.9)	14 (35.9)	12 (12.6)	15 (17.9)	11 (22.4)	16 (25)	11 (15.9)	7 (15.3)	8 (40)	5 (38.4)
*p*-value	0.048 *		0.296		<0.001 *		0.900		0.003 *		0.753	
*V-Cramer*	0.153		0.172		0.271		0.0433		0.3166		0.131	
Reworking social media photos. n (%)												
Yes, always	34 (24.6)	38 (30.9)	18 (41.9)	18 (46.2)	16 (16.8)	20 (23.8)	16 (32.7)	23 (35.9)	13 (18.8)	11 (23.9)	5 (25)	4 (30.8)
Yes, sometimes	48 (34.8)	58 (47.2)	8 (18.6)	14 (35.9)	40 (42.1)	44 (52.4)	18 (36.7)	25 (39.1)	27 (39.1)	27 (58.7)	3 (15)	6 (46.2)
No, never	56 (40.6)	27 (22)	17 (39.5)	7 (17.9)	39 (41.1)	20 (23.8)	15 (30.6)	16 (25)	29 (42)	8 (17.4)	12 (60)	3 (23.1)
*p*-value	0.005 *		0.060		0.047 *		0.801		0.020 *		0.072	
*V-Cramer*	0.200		0.262		0.185		0.063		0.260		0.399	

HSS—high school students; CS—college student; WP—working people; * *p* < 0.05.

**Table 6 nutrients-17-00219-t006:** Linear regression analysis of MEH score and EAT-26 score in sex group (n = 261).

Model Coefficients—EAT-26; R = 0.339 R^2^ = 0.115				
Predictor	Estimate	SE	t	*p*-Value
MEH Total	0.429	0.090	4.746	<0.001
Factor I	0.0158	0.267	0.0592	0.837
Factor II	0.4239	0.305	1.3896	0.182
Factor III	0.9239	0.254	3.6437	<0.001
Gender M-F	−0.0542	1.307	−0.0414	0.967

M—male; F—female; EAT-26—Eating Attitudes Test-26; MEH—My Eating Habits Questionnaire; SE—standard error; t—t-statistics.

**Table 7 nutrients-17-00219-t007:** Linear regression analysis of EAT-26 and EAT-26 subscale score in gender group (n = 261).

Model Coefficients—EAT-26; R = 0.971 R^2^ = 0.943				
Predictor	Estimate	SE	t	*p*-Value
Dieting	1.20	0.0277	43.20	<0.001 *
Bulimia	1.18	0.0623	18.90	<0.001 *
Oral control	1.17	0.0603	19.44	<0.001 *
Gender M-F	−0.9619	0.3114	−3.0894	0.002 *

M—male; F—female; EAT-26—Eating Attitudes Test-26; SE—standard error; t—t-statistics; * = *p* < 0.05.

**Table 8 nutrients-17-00219-t008:** Linear regression analysis of EAT-26 and SES in sex group (n = 261).

Model Coefficients—EAT-26; R = 0.114 R^2^ = 0.013				
Predictor	Estimate	SE	t	*p*-Value
SES	0.00503	0.177	0.02984	0.977
Gender M-F	0.00503	0.177	0.0284	0.070

M—male; F—female; EAT-26—Eating Attitudes Test-26; SES—Rosenberg Self-Esteem Scale; SE—standard error; t—t-statistics.

**Table 9 nutrients-17-00219-t009:** Analysis of Covariance (ANCOVA) for EAT-26 Scores—Examining the Influence of Gender and Time Spent on Social Media.

ANCOVA—EAT 26 Score					
	Sum of Squares	df	Mean Squares	F	*p*-Value
Gender	148.0	1	148.00	1.6774	0.196
How much time do you spend daily on social media?	17.5	3	5.84	0.0662	0.978
Gender vs. Time spent daily on MS	399.4	3	133.14	1.5090	0.213
Residuals	22,322.3	253	88.23		

## Data Availability

The raw data supporting the conclusions of this article will be made available by the authors upon request.

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
