# Peer review of "The Impact of Social Media on Eating Disorder Risk and Self-Esteem Among Adolescents and Young Adults: A Psychosocial Analysis in Individuals Aged 16–25"

_nutrients, 2025, doi:10.3390/nu17020219_

Round 1

Reviewer 1 Report

Comments and Suggestions for Authors

Evaluation of manuscript nutrients-3416460

This manuscript performed an important analysis about ED in subjects between 16-25 yrs. I will present below my considerations about the text aiming to help the authors to improve the quality of the text.

The abstract is very complete, but I missed observing numerical results alongside the statistics, please review.

The introduction is well written, however, the way the authors write it, it seems to me that the present study is a mere repetition of others previously published. After all, what are the original aspects of this manuscript. The authors talk about the relevance of the subject, and this is well done, however, what does the present study advance in relation to previously published studies? Without plausible justification, I will oppose the publication of this manuscript.

In the methods it is important that we have a calculation of sample representativeness. After all, n = 275 may seem like a large number for a certain type of inferential analysis, but it may be small for a certain type of experimental design.

The details of BMI calculations are unnecessary, as it is a classic and widely known method. The most interesting thing is to talk about the cutoff point adopted.

Please, add the 95% CI in Table 1 and 4.

As for statistical methods, it is not necessary to perform a normality test for Fisher's exact or Chi-square, this type of test is non-parametric. The normality test will be necessary for the regression, please review the wording.

Please enter the effect size for the chi-square (Cramer’s V) next to each statistical result.

All Tables must be self-explanatory. Some acronyms are not explained in the Table note, e.g. SES, MEH, please review.

I consider the discussion to be the weakest point of the study, the authors carry out a literature review and do not explain the results. I suggest that you rewrite, for each result, or set of results of the present study, a logical explanation must be presented based on previously published studies. Please review.

Author Response

Thank you so much for taking the time to evaluate our work. We have tried to incorporate all your valuable suggestions. If we could improve our work in any way, please let us know.

Comment 1

Abstract:

The abstract is very complete, but I missed observing numerical results alongside the statistics, please review.

Thank you very much for your suggestion. The results section in the abstract was completed.

Comment 2

Introduction

The introduction is well written, however, the way the authors write it, it seems to me that the present study is a mere repetition of others previously published. After all, what are the original aspects of this manuscript. The authors talk about the relevance of the subject, and this is well done, however, what does the present study advance in relation to previously published studies? Without plausible justification, I will oppose the publication of this manuscript.

Thank you very much for your suggestions, we have corrected the introduction and added the relevant section (lines 73-84). In addition, the discussion section has been revised and emphasizes why this type of research is important, even though the topic of eating disorders is common in scientific research. One reason is the evolution of risk factors. Modern technologies, such as social media, and their impact on mental health, including self-esteem and body image, are relatively new aspects that need attention. Eating disorder research must adapt to these changes in order to effectively identify new risk factors and adapt prevention strategies. Conducting research on different populations (a study on Polish adolescent and adult populations) allows us to better understand these differences and develop more effective, personalized therapeutic approaches.

Comment 3

In the methods it is important that we have a calculation of sample representativeness. After all, n = 275 may seem like a large number for a certain type of inferential analysis, but it may be small for a certain type of experimental design.

Thank you very much for your suggestion, corrected (section 2.2).

Comment 4

The details of BMI calculations are unnecessary, as it is a classic and widely known method. The most interesting thing is to talk about the cutoff point adopted.

Thank you very much for your suggestion, corrected.

Comment 5

Please, add the 95% CI in Table 1 and 4.

Thank you very much for your valuable comment, Table 1 and 4 contain the confidence interval.

Comment 6

As for statistical methods, it is not necessary to perform a normality test for Fisher's exact or Chi-square, this type of test is non-parametric. The normality test will be necessary for the regression, please review the wording.

Thank you very much for your suggestion, corrected.

Comment 7

Please enter the effect size for the chi-square (Cramer’s V) next to each statistical result.

Thank you very much for your valuable comment, corrected.

Comment 8

All Tables must be self-explanatory. Some acronyms are not explained in the Table note, e.g. SES, MEH, please review.

Thank you very much for your suggestion, corrected.

Comment 9

I consider the discussion to be the weakest point of the study, the authors carry out a literature review and do not explain the results. I suggest that you rewrite, for each result, or set of results of the present study, a logical explanation must be presented based on previously published studies. Please review.

Thank you for bringing this to our attention. We agree with this comment, so the discussion section has been corrected as suggested by the reviewer.

Revised the manuxcript to meet expectations.

Thank you for your help. Your guidance is invaluable.

Kind regards,

Authors.

Reviewer 2 Report

Comments and Suggestions for Authors

Thank you for giving me the opportunity to review this paper, which represents a significant contribution to understanding the impact of social media on the risk of eating disorders and self-esteem among adolescents and young adults. The topic is relevant and well-structured, with important implications for the mental health and well-being of young people. Below, I provide my comments to further enhance your work.

Strengths

  1. Title and Abstract: The title is clear and engaging. The abstract effectively summarizes the objectives, methods, results, and implications of the study.
  2. Methodology: The choice of validated tools, such as the EAT-26 and the Rosenberg Self-Esteem Scale, adds robustness to the study. The CAWI method is well-justified.
  3. Discussion: It connects the findings with the existing literature, providing valuable insights for future interventions.
  4. Statistical Analysis: The presentation of group differences, the role of photo manipulation on social media, and the impact of body comparisons is well-articulated and supported by strong evidence.

Suggestions for Improvement

  1. Abstract: Consider incorporating a brief mention of the study's limitations and possible directions for future research.
  2. Introduction:
    • It would be beneficial to cite Diotaiuti et al. (2022) to support the use of psychometric tools. This could be included in the section where the importance of the scales used to measure eating behaviors is discussed (e.g., after the description of the EAT-26).
      • Diotaiuti, P., et al. (2022). Psychometric properties and measurement invariance across gender of the Italian version of the tempest self-regulation questionnaire for eating adapted for young adults. Frontiers in psychology, 13, 941784. https://doi.org/10.3389/fpsyg.2022.941784.
  3. Methodology:
    • Provide more details on the statistical power of the sample and justify the exclusion of some sections of the EAT-26 (line 120). This would enhance methodological clarity.
    • Consider summarizing the details regarding ethical approval (lines 84-92) to avoid redundancy.
  4. Results:
    • Add graphical representations or tables to better visualize key correlations and significant differences between groups.
    • Further explore subgroups, detailing the interactions between social media usage time, gender, and ED risk.
  5. Discussion:
    • You could integrate the article by Cavicchiolo et al. (2022) in the final section to strengthen your recommendations for educational and psychological interventions for young adults.
      • Cavicchiolo, E., et al. (2022). The Psychometric Properties of the Behavioural Regulation in Exercise Questionnaire (BREQ-3): Factorial Structure, Invariance and Validity in the Italian Context. International journal of environmental research and public health, 19(4), 1937. https://doi.org/10.3390/ijerph19041937.

Sections to Review or Remove

  1. Methodology (lines 84-92): The information on ethical approval is important but can be condensed.
  2. Discussion (lines 375-378): The educational implications already mentioned in the results could be reduced to avoid repetition.

Study Limitations

I suggest highlighting:

  • The reliance on self-reported data, which could introduce bias.
  • The lack of objective measures (e.g., physiological or behavioral analyses) to complement the findings.
  • The exclusive focus on the Polish population, which might limit the generalizability of the results

Author Response

Thank you so much for taking the time to evaluate our work. We have tried to incorporate all your valuable suggestions. If we could improve our work in any way, please let us know.

Comment 1

Abstract: 

Consider incorporating a brief mention of the study's limitations and possible directions for future research

Thank you very much for your suggestion, corrected (lines 29-31).

Comment 2

Introduction:

It would be beneficial to cite Diotaiuti et al. (2022) to support the use of psychometric tools. This could be included in the section where the importance of the scales used to measure eating behaviors is discussed (e.g., after the description of the EAT-26).

Thank you for your valuable suggestion, updated.

Comment 3

Methodology:

Provide more details on the statistical power of the sample and justify the exclusion of some sections of the EAT-26 (line 120). This would enhance methodological clarity.

Thank you very much for your suggestion, corrected (lines104-110 and lines 148-153).

Consider summarizing the details regarding ethical approval (lines 84-92) to avoid redundancy.

Thank you very much for your suggestion, corrected.

Comment 4

Results:

Add graphical representations or tables to better visualize key correlations and significant differences between groups.

Further explore subgroups, detailing the interactions between social media usage time, gender, and ED risk.

In response to the reviewer's comments, graphical representations has been added. Additionally, subgroup analyses have been conducted to explore the interactions between social media usage time, gender, and ED risk, providing a more detailed breakdown and enhancing the comprehensiveness of the findings.

Comment 5

Discussion:

You could integrate the article by Cavicchiolo et al. (2022) in the final section to strengthen your recommendations for educational and psychological interventions for young adults.

Thank you for your valuable suggestion, updated.

Comment 6

Sections to Review or Remove

Methodology (lines 84-92): The information on ethical approval is important but can be condensed.

Thank you very much for your suggestion, corrected.

Discussion (lines 375-378): The educational implications already mentioned in the results could be reduced to avoid repetition.

Thank you very much for your suggestion, corrected.

Comment 7

Study Limitations

I suggest highlighting:

  • The reliance on self-reported data, which could introduce bias.
  • The lack of objective measures (e.g., physiological or behavioral analyses) to complement the findings.
  • The exclusive focus on the Polish population, which might limit the generalizability of the results

Thank you very much for your suggestion, corrected.

Thank you for your valuable feedback and for taking the time to review our manuscript titled: " The Impact of Social Media on Eating Disorder Risk and Self-Esteem Among Adolescents and Young Adults: A Psychosocial Analysis in Individuals Aged 16-25."

We appreciate your constructive comments and have carefully revised the manuscript to address each point you raised. The following adjustments have been made:
We have thoroughly revised the manuscript to improve clarity, coherence, and overall readability. Sections that were previously repeated parts or unclear, particularly within the methods, have been rewritten to ensure transparency and precision.

The methodology section has been expanded to provide a clearer description of the calculation of sample representativeness.

In results has been added graphical representations to better visualize key correlations and significant differences between groups.

We sincerely hope that these revisions address the concerns raised and enhance the quality and rigor of our manuscript. We believe the revised version aligns more closely with the objectives of the study and the standards expected for publication

Thank you once again for your valuable insights. We appreciate your time and effort in helping us improve our work and look forward to receiving further feedback.

Kind regards,

Authors

Round 2

Reviewer 1 Report

Comments and Suggestions for Authors

The authors' efforts to improve the quality of the manuscript can be seen, especially regarding the representativeness of the sample and instruments, which was a point that I criticized in the first version. There is a great improvement in the current manuscript and I am in favor of publishing it as it stands.

Reviewer 2 Report

Comments and Suggestions for Authors

After a careful review of the manuscript and the modifications the authors have implemented following our suggestions, we are pleased to inform you that the article has now reached a level of quality and completeness that meets the requirements for publication.

The proposed revisions have been effectively incorporated, enhancing the clarity, coherence, and depth of the content. The structure of the article is now solid and well-organized, with arguments thoroughly developed and supported by appropriate evidence. Additionally, the authors have diligently addressed the points raised during the review process, demonstrating particular attention to scientific accuracy and clarity of presentation.

Given the results achieved and the high quality of the content, we believe the article is now ready for publication. We are confident that this work will make a significant contribution to the literature in the field and will be of great interest to the scientific community.

Best regards